# Evolving Strategies for Use of Phytochemicals in Prevention and Long-Term Management of Cardiovascular Diseases (CVD)

**DOI:** 10.3390/ijms25116176

**Published:** 2024-06-04

**Authors:** Donald David Haines, Fred M. Cowan, Arpad Tosaki

**Affiliations:** 1Advanced Biotherapeutics Ltd., 20-22, Wenlock Road, London N1 7GU, UK; 2Uppsala Inc., 67 Shady Brook Drive, Colora, MD 21917, USA; fred.cowan@zoominternet.net; 3Department Pharmacology, Faculty of Pharmacy, University of Debrecen, Nagyerdei krt. 98, 4032 Debrecen, Hungary; 4HUN-REN-UD Pharmamodul Research Group, University of Debrecen, Nagyerdei krt. 98, 4032 Debrecen, Hungary

**Keywords:** inflammation, cardiovascular disease, ischemia, spice, phytochemicals, *Ginkgo biloba*, sour cherry, heme oxygenase, interleukin-8, screening

## Abstract

This report describes major pathomechanisms of disease in which the dysregulation of host inflammatory processes is a major factor, with cardiovascular disease (CVD) as a primary model, and reviews strategies for countermeasures based on synergistic interaction between various agents, including drugs and generally regarded as safe (GRAS) natural medical material (NMM), such as *Ginkgo biloba*, spice phytochemicals, and fruit seed flavonoids. The 15 well-defined CVD classes are explored with particular emphasis on the extent to which oxidative stressors and associated ischemia-reperfusion tissue injury contribute to major symptoms. The four major categories of pharmaceutical agents used for the prevention of and therapy for CVD: statins, beta blockers (β-blockers), blood thinners (anticoagulants), and aspirin, are presented along with their adverse effects. Analyses of major cellular and molecular features of drug- and NMM-mediated cardioprotective processes are provided in the context of their development for human clinical application. Future directions of the evolving research described here will be particularly focused on the characterization and manipulation of calcium- and calcineurin-mediated cascades of signaling from cell surface receptors on cardiovascular and immune cells to the nucleus, with the emergence of both protective and pathological epigenetic features that may be modulated by synergistically-acting combinations of drugs and phytochemicals in which phytochemicals interact with cells to promote signaling that reduces the effective dosage and thus (often) toxicity of drugs.

## 1. Introduction and Main Topic Review

Inflammation-associated oxidative stress and tissue injury: Dysregulated inflammation is a driving mechanism underlying the major symptoms of cardiovascular disease (CVD) and pathologenesis of many illnesses. Pathological inflammation is currently treated with small molecule pharmaceutical agents that are typically designed to act at single critical checkpoints in the inflammatory cascade. This strategy may be effective for suppressing major symptoms of many diseases. However, inflammatory processes are rarely cured by anti-inflammatory drugs presently available for human use and long-term treatments are often limited by drug cost and toxicity. Moreover, many phytochemicals widely distributed in human diet, particularly in common spices, have anti-inflammatory properties but are individually non- or marginally therapeutic. Previously, the authors of this report and others have shown that multiple phytochemicals may interact collectively to achieve efficacy similar to pharmaceutical drugs [1,2,3,4]. A major limitation of using phytochemicals in healthcare is that most of these compounds individually have very low potencies in comparison to anti-inflammatory drugs currently in use and require high doses to achieve even modest efficacy. These limitations can, however, be overcome by the use of formulations more reflective of a healthy diet, containing multiple phytochemicals that interact collectively. This phenomenon, called “phytochemical synergy” is a major concept considered in the current study. Spice-derived phytochemicals are an excellent example of this paradigm for inhibiting inflammatory tissue damage manifesting as disease symptoms. These dietary materials have been shown to constrain pro-inflammatory signaling mechanisms such as redox reactions and the nuclear factor-kappa beta (NF-kB) pathway to their appropriate physiologic venues—offering promise for the management of pathological inflammation [5,6].

Spice flavorings and nuclear-factor kappa beta Processes that involve NF-kB have emerged as very important mediators of disease- or trauma-associated inflammation, particularly as a major characteristic of syndromes with increased incidence among elderly populations such as cancer, atherosclerosis, cardiovascular disorders, diabetes, allergies, asthma, Alzheimer’s disease, osteoporosis, and many forms of autoimmunity [5,6].

The physiological and pharmacological mechanisms by which phytochemicals affect pro-inflammatory signaling pathways are an increasing focus of cutting-edge nutritional science. A case example of how this research may allow increasing use of phytochemicals in preventive medicine and therapy is provided by the characterization of SIRT1, a histone deacetylase (HDAC) enzyme upregulated by stress and also by certain compounds found in the human diet, such resveratrol, a phytochemical produced by grapes. SIRT1 directly binds to and regulates molecules of the NF-kB cascade [7,8], and may also impact the epigenomic profile in cells that express this enzyme. One specific outcome of this activity is that resveratrol can act through SIRT1 to inhibit NF-kB, resulting in the enhancement of tumor necrosis factor-alpha (TNF-A)-mediated apoptosis [7,8]. Apoptosis, a cell death program that removes damaged or dysregulated inflammatory cells, is a major mechanism for down-regulating inflammation and preventing it from becoming pathological [9,10].

Resveratrol analogs are being developed by Sirtris Pharmaceuticals partnered with GlaxoSmithKline as proprietary, orally available, small molecule drugs with the potential to treat age-associated diseases, such as type 2 diabetes, through the modulation of sirtuin activity. These efforts by Sirtris focus on modulating the expression of sirtuins. Curcumin, the major active phytochemical component of the spice turmeric, has shown some efficacy against inflammatory and neoplastic disorders—including pancreatic cancer, which is notoriously resistant to therapy [11,12]. Curcumin is also the subject of research to develop more potent analogs with increased bioavailability [5,6]. Other examples of therapeutic application of dietary phytochemicals include the use of preparations extracted from red pepper as topical analgesics and anti-inflammatories. Potent analogs of compounds found in pepper, such as capsaicin, have also been developed for medical use. Nevertheless, synthetic phytochemical analogs have the disadvantage of high cost and potential for adverse side effects that are avoided by naturally occurring bioactive compounds. A major objective in the development of phytochemicals for use in healthcare is to identify combinations of common foods and flavorings that synergize in strengthening immunoregulation.

Recent reports show the validity of phytochemical synergy as a healthcare strategy. For example, green tea concentrate and preparations of vanilloid compounds obtained from capsicum (red pepper) exhibit anti-cancer properties suggestive of synergy between polyphenols in tea called catechins and capsicum vanilloids, resulting in the inhibition of tumor-associated NADH oxidase (tNOX), an enzyme contributing to cancer proliferation [13,14]. Other studies demonstrated that combinations of curcumin and resveratrol exhibit 4-fold antioxidant synergism in heme-enhanced oxidation reactions [15,16]. Curcumin and resveratrol have also been observed to synergistically inhibit IL-8 expression by human cells in vitro exposed to the chemical weapon sulfur mustard (Cowan et al. unpublished observations). An area of special interest but limited experimentation and information is the ability of phytochemicals to augment the therapeutic potential of pharmaceutical drugs by enhancing their potency, which allows a reduction in drug dose, thus lessening toxicity [9,10,17,18,19]. An example of this effect is provided by the use of *Ginkgo biloba* extracts to lower the effective dosage of the immunosuppressive drug, FK506, required for cardioprotection in vivo [18,20]. Related work with ginkgolide B (GB), a component molecule of *Ginkgo biloba* extract, reduces the dosage of the immunosuppressant drug cyclosporin A required to inhibit inflammatory cytokine expression by human T cells in vitro [3,6]. Ginkgo leaf extract has also been shown to interact with vitamin C and the antioxidant carotenoid astaxanthin to mediate anti-inflammatory effects with the same potency as Ibuprofen, but without risk of the adverse gastrointestinal side effects caused by Ibuprofen and other NSAIDs [21,22]. These and related studies support the hypothesis that components of phytochemical-rich food and herbal products can interact to synergistically improve the efficacy of drug therapies [20].

Some of the biochemical, physiological, and pharmacological mechanisms mediated by phytochemicals have been reviewed; examples of this work include “Reasoning for Seasoning” [5], which characterized links between pro-inflammatory cellular signaling pathways and the capacity of spice-derived phytochemicals to inhibit NF-kB-mediated inflammatory processes [23].

Strategies for managing cancer using combinations of multiple phytochemicals (“Multifocal Signal Modulation Therapy”) were proposed by McCarty (2021) [24], and a comprehensive review of how the pro-apoptotic action of NF-kB-inhibitory phytochemicals may act as a pivotal defense against pathological inflammation and disease is described in Martins (2006) [10].

Previous and ongoing investigations suggest a major role for inflammation in chemical toxicity and support the potential efficacy of anti-inflammatory drugs and phytochemicals against chemical warfare agents (reviewed by Cowan et al., 2004, 2006, and unpublished observations); and: [19,25,26].

This work was the basis for Multi-Threat Medical Countermeasures (MTMC) technology which aimed to develop treatments for pathologies triggered by multiple classes of chemical agents with a single drug protocol (Cowan et al., 2004, 2006, and unpublished observations) [19,25,26,27]. MTMC technology may also find more general public heath applications because of the commonality of inflammatory mechanisms in chemical insult and disease [26,28,29]. MTMC synergistic combinations of anti-inflammatory drugs and/or phytochemicals have been confirmed in assay and animal models [29,30]. Further drugs with MTMC primary or pleotropic anti-inflammatory pharmacology are prescribed and administered to millions of patients, often in polypharmacy, with potential synergy and implications for the design of MTMC polypills [30]. MTMC can be a generic drug or supplements, thereby addressing affordability, which are practiced in any small clinic, reducing R&D costs. MTMC synergy assays and animal models for inflammation are readily available [26,31]. GLP-1 diet drugs are FDA approved for cardiovascular disease and have demonstrated efficacy for myriad other inflammatory pathologies [32]. Concerns of cost for on-patent MTMC vs. more affordable generic MTMC have been considered [28].

IL-8 is produced at the end of NF-kB-mediated cytokine cascades and its level in peripheral blood is a particularly reliable biomarker of pathological inflammation and the efficacy of anti-inflammatory therapies. IL-8 is a chemotactic, which strongly recruits neutrophils and inhibits their apoptosis, thereby impairing the resolution of inflammation and establishing conditions for chronic disease [5,6,10]. In previous studies, elevated levels of IL-8 have been associated with insult, injury, and a variety of pathologies and disease; these include chemical insult, specifically sulfur mustard exposure, and acute respiratory distress syndrome (ARDS) and associated multi-organ dysfunction syndrome (MODS) [25,26,33,34]. Sulfur mustard-increased IL-8 is a biomarker for the severity of mustard-induced inflammatory pathology and may also be used to gauge the efficacy of medical countermeasures to this chemical weapon [19,25,26].

IL-8, along with CRP and the pro-inflammatory cytokine interleukin-6 (IL-6), is an indicator of pulmonary disease severity in mustard-exposed persons [35,36,37,38].

The phytochemical capsaicin and its more potent drug analogs inhibit sulfur mustard-increased IL-8 production in cell culture and show efficacy against sulfur mustard-associated pathology in the mouse ear vesicant model [19,26,39]. Compounds that inhibit IL-8 production and reverse the anti-apoptotic and pro-inflammatory effects of this cytokine include many phytochemicals such as capsicum, curcumin, and resveratrol [5,10,25]. Such apoptotic pharmacology is also commonly observed for phytochemicals and drugs used as chemical agent countermeasures [10,25]. The proven capacity of chemical agents to trigger inflammatory processes and the role of inflammation in a wide range of pathologies makes phytochemical synergy fertile ground for research [19,40].

The capacity of numerous phytochemicals to inhibit IL-8 expression and associated inflammation makes this cytokine a logical biomarker for measuring LPS-increased inflammatory response by PBMC in the present study. The data on Aqua Spice™ food flavorings presented in this report advance the concept of phytochemical synergy. Phytochemical synergy technology is being used to design ImmuneACCORD™ dietary supplements. ImmuneACCORD Technology which encompasses the process of screening for phytochemical synergy, seeks to define mechanistic links between diet, dietary supplements, and drug discovery with the goal of more affordable and improved public health.

## 2. Cardiovascular Diseases (CVDs) and Ischemia-Reperfusion: A Paradigm of Oxidative Stress-Mediated Pathogenesis

Cardiovascular diseases are a category of syndromes involving acute and progressive damage to vascular and heart tissues which are classed into 15 well-defined disease processes, including venous thrombosis, heart failure, stroke, myocardial infarction (heart attack), coronary artery diseases (CADs), such as carditis, angina, and ventricular fibrillation, which are responsible for sudden cardiac death [41], hypertensive heart disease, rheumatic heart disease, peripheral arterial disease, valvular heart disease, congenital heart disease, thromboembolic disease, aortic aneurysms, and cardiomyopathy [42,43]. The phenomenon of dysregulated inflammation occurs as a prominent aspect of the 15th class of CVD on this list, ischemic/reperfusion (I/R)-associated disease [4,42,43,44,45], which is a major focus of the present report. The mitigation of these syndromes with synergistically acting phytochemicals present in functional foods is the clinical objective of the in vitro animal and human clinical strategies described here. The experiments described in the studies described in this review were conducted to assess combinations of drugs and natural products, in particular, ‘functional foods’ such as sour cherry extracts and spices for synergism, in suppressing the major pro-inflammatory mechanisms contributing to CVD and all other diseases in which these processes are contributors to pathogensis and symptom severity. CVD is here presented as a representative paradigm of disease in which hyperactivated host inflammation is a primary driver of emergence and progression of major symptoms. It is further important to emphasize that since this phenomenon is common to almost all known disorders, the outcomes of this work are broadly applicable for an enormous number of clinical venues. The findings revealed by the investigations described in this report are particularly exciting since they suggest a means of expanding and refining the known health benefits of foodstuffs that are widely available at low cost. Particularly exciting are reports showing the strong ability of commonly used spice flavorings to reduce the severity of cardiovascular ischemia/reperfusion-related symptoms, notably, turmeric [46], ginger [47,48], capsicum/capsaicin—in red pepper [49,50]—including paprika [51,52] and nutmeg [53,54,55]. Each of these spices contain bioactive compounds that interact to exhibit very potent anti-inflammatory synergy, with high potential for the long-term clinical management of CVD and many other disorders at minimal cost and with no adverse side effects.

Within the past half century, cardiovascular disorders have emerged as the primary cause of mortality worldwide, except among African populations, with the highest rates observed among residents of affluent nations, whose lifestyle-associated risk factors, notably fat-rich diets and general ‘overnutrition’, heavy tobacco and alcohol use, and lack of exercise, are typically higher than among people of developing countries [42,44]. Additionally, some strains of pathogenic streptococci responsible for strep throat may trigger host autoreactive attack on cardiovascular tissue, resulting in rheumatic heart disease following long latency periods if the primary infection is not treated with antibiotics at the time of occurrence [42,44].

The toll of these disorders is striking and has substantially increased in recent years. By 2015, 17.9 million deaths among human populations were attributable annually to cardiovascular diseases, which is an increase of 25.8% (12.3 million) from the number recorded in 1990 [43,56] and continued increases have been noted recently [4,42,57]. It is nevertheless interesting to note that, during the past decade, the personal utilization of risk reduction strategies available through public education has resulted in some abatement of mortality rates in affluent societies. This includes, prominently, an awareness of the benefits to general health of the dietary intake of polyphenols, especially in flavonoid-rich foods, and flavorings such as the spices described in the present report [5,6,46,47,48,49,50,51,52,53,54,55,56,57,58,59,60].

The data shown here and an analysis of the work by other investigators underscores this trend in public health consciousness and demonstrates how certain spice combinations may exert additive and, in some cases synergistic capacity for the prevention and long-term, side effect-free, management of CVDs, along with a diverse range of other disorders in which major symptoms result from hyperinflammatory reactions.

The underlying pathophysiological mechanisms for each of the syndromes listed above have been extensively documented. Peripheral and coronary artery disease and stroke, often progressing concomitantly with atherosclerotic changes to the vasculature, show strong correlations with sleep disorders and related psychological influences [61,62,63], lack of exercise, alcohol and tobacco use, elevated blood cholesterol, obesity with insulin resistance and resulting type 2 diabetes, obesity, and high blood pressure [42,44]. Significantly, in the context of findings reported here, approximately 53% of CVD-associated mortality is currently linked to dietary risk factors [64].

Ultrastructural alterations typically observed in the heart and vascular tissue of CVD patients that correlate with deteriorated function, typically include fibrotic changes and the accumulation of amyloid deposits in the interstitium—as shown by the immunohistochemical micrograph in Figure 1. Inflammatory heart diseases are particularly debilitating, refractory to treatment (especially long-term), and often fatal [44,65]. Disorders in this category include such conditions as endocarditis, in which the heart’s inner layer, the endocardium, becomes inflamed, with resulting damage to the heart valves [66]. Likewise, cardiomegaly, a disorder characterized by the hypertrophic enlargement of the organ, with resulting loss of function, and myocarditis, a disease process in which monocytes and lymphocytes or eosinophiles infiltrate the heart muscle, are consequences of a failure to regulate inflammatory signaling processes, with consequent tissue damage [67,68,69,70,71,72]. Significantly, the triggering etiology for these pathologies may be viral, bacterial, or toxin-induced [69,70], thus raising the risk that some infectious diseases may engender cardiovascular complications.

A major positive aspect to the current understanding of these diseases is that it is estimated that as much as 90% of CVD may be prevented by intuitively sensible lifestyle measures, especially an avoidance of alcohol and tobacco use, along with exercise and a healthy diet rich in foods containing antioxidant polyphenols [73,74,75]. It has also been observed that patient compliance with regimens for the therapy of CVD risk factors such as diabetes, high blood pressure, and elevated blood lipids, also has significant corollary effects in CVD prevention and treatment [73,74,75].

An increasingly broad and versatile selection of drugs for the prevention and treatment of CVD has come into clinical use in recent years. A detailed consideration of the full range of emerging CVD drugs is beyond the scope of the present report. Thus, in order to clearly present how phytochemical synergy may be used as a countermeasure to the onset and progression of the driving inflammatory pathology underlying CVD—and indeed, all other diseases at some level—the major advantages and drawbacks to four of the best-known representatives of CVD pharmacotherapies will be considered here, with these being statins, beta blockers, blood thinners, and aspirin.

## 3. Representative Pharmaceutical Agents Used for CVD

### 3.1. Statins

These are a class of agents widely used for the prevention and management of CVD which function through the inhibition of the enzyme 3-hydroxy-3-methyl-glutaryl-coenzyme A (HMG-CoA) reductase, which in mammalian cells, including humans, is the major NADH-dependent regulator of mevalonate metabolism and processes responsible for the biosynthesis of isoprenoids, including cholesterol. In healthy cells, the activity of this enzyme is competitively suppressed through negative feedback thus limiting the availability of cholesterol to levels of optimal use in healthy physiologic function [76,77]. Statins inhibit HMG-CoA reductase, thereby reducing cholesterol production at levels that, depending on the overall health of an individual and the presence of other risk factors, may decrease the development of dyslipidemia and the resulting illness and death [76,78].

Adverse effects of statins. The statin-mediated suppression of cholesterol production is accompanied by a concomitant reduction in the blood levels of low-density lipoprotein (LDL) cholesterol, a complex of cholesterol with a carrier protein, responsible for the transport of fats to cells. Most cholesterol in a body is bound in this form. Elevated LDL levels correlate with an increased risk of stroke, heart disease, and related problems, and influences that reduce LDL also decrease the risk of a wide range of cardiovascular problems [79,80]. The LDL-lowering properties of statin drugs have thus made this class of compounds a highly potent clinical countermeasure to CVDs; nevertheless, their use is also associated with a number of significantly adverse side effects such as aberrant liver enzyme activity in peripheral blood and an elevated risk for type 2 diabetes [81,82], along with a rare but severe occurrence of muscle impairment and pain [83]—and anecdotal but increasingly frequent reports of hepatotoxicity.

### 3.2. Beta Blockers (β-Blockers)

This class of drugs is a compound that competitively antagonizes the binding of the endogenous catecholamines, adrenaline (epinephrine) and noradrenaline (norepinephrine), to adrenergic beta receptors which are their cognate sites expressed by cells of the sympathetic nervous system and by major organs participating in stress responses, notably, downstream events such as fight-or-flight response [84,85]. This class of drugs is found to be particularly valuable in stabilizing cardiac arrhythmias to reduce the occurrence of secondary heart attacks following a first attack, and are also useful in the treatment of high blood pressure [85,86]. Three variants of β-adrenergic receptors are known (β1, β2, and β3), each with characteristic anatomical distribution to major organs such as the heart, lungs, uterus, smooth and skeletal muscle, kidneys, gastrointestinal tract, and vascular and adipose cells [84,85]. Some beta blockers are capable of blocking the activation of all three of these receptors, while others exhibit selectivity for one receptor type [85,87]. The stimulation of beta receptors with adrenaline on components of the sympathetic nervous system, especially within the kidneys, smooth muscle, heart muscle, arteries, and tracheobronchal tree, produce stress responses that may be deleterious to the heath of an affected individual. Consequently, beta blockers that inhibit the binding of this hormone to its complementary beta receptor may attenuate its adverse effects. Indeed, these drugs have proven very useful since the pioneering work of James Black in 1964 showing the potent capacity of the beta receptor-blocking drugs, pronethalol and propranolol, in the treatment of angina pectoris, which is considered to be one of the of the 20th century’s most significant developments in clinical pharmacology [88]. The sympatholytic β1 activity of beta blockers decrease incidence and severity of congestive heart failure—and further suppress pathological inflammatory activity through effects on the renin–angiotensin system [85,89]. The interaction of these drugs with their receptors in the kidneys reduces renin secretion, causing a principal effect of suppressed oxygen requirements of the heart through decreased extracellular fluid volume, thereby engendering concomitant increased capacity of the blood to carry oxygen [89]. The notable beneficial consequence to this on the prognosis of heart failure and associated inflammatory tissue damage arises from the inhibition of catecholamine influences on the heart, a phenomenon shown to be responsible for the elevated expression of reactive oxygen species (ROS) and other mediators of inflammation such as the cytokines TNF-A, interleukin-1beta (IL-1B), and interleukin-6 (IL-6)—in addition to the catecholamine-mediated deleterious remodeling of cardiac tissue [90,91], high oxygen demand, and decreased cardiac contractile efficiency—effects which collectively decrease the ejection fraction volume and exacerbate inflammatory damage to the heart [91]. These very potent effects have resulted in substantial benefits for the prevention and management of CVD. Indeed, by 2004, the use of beta blockers was shown to be primarily responsible for a 4.5% reduction in mortality during a 13-month representative timeframe in which clinical evaluations were conducted [92].

Adverse effects of beta blockers. The clinical advantages of beta blockers notwithstanding, these drugs are also associated with significantly serious adverse side effects. For example, this repertoire of problematic outcomes includes the results of a 2007 investigation indicating that their use when co-administered with diuretics for hypertension increase the risk for type 2 diabetes—which may be counteracted, to some extent, by combining such a regimen with angiotensin II receptor blockers [93]. Moreover, lipophilic beta blockers, including the commonly used drugs propranolol and metoprolol, which have high penetration across the blood–brain barrier, are known to result in sleep disturbances, including insomnia and nightmares [94].

Further disruption of healthy physiologic processes contributing to metabolic syndrome may occur as a result of the inhibition by these drugs of β2-adrenoceptor-mediated glycogenolysis by the liver and subsequent pancreatic glucagon release, which normally increases plasma glucose as needed, but, when inhibited, may lead to hypoglycemia [93].

Other beta blocker-associated side effects include exacerbation of Raynaud’s syndrome, cold extremities, hypotension, bradycardia, heart failure, dizziness and fatigue, bronchospasm, peripheral vasoconstriction, hair loss, insomnia, nightmares and hallucinations, insomnia, erectile dysfunction, diarrhea, dyspnea, blurred vision, and nausea [94,95]. The above-described complications and many other adverse effects make this class of drugs very much a double-edged sword for the prevention of and therapy for CVD.

### 3.3. Blood Thinners (Anticoagulants)

Pharmaceutical agents in this class are anticoagulants, which interfere with the normal cascade of biochemical events leading to clot formation—a drug-mediated effect that extends clotting time and decreases blood viscosity, thereby reducing the risk of thrombus formation and stroke [96,97]. These drugs may have both a direct and corollary regulatory effect on dysregulated inflammation in ischemia/reperfusion-associated pathologies, through relief of pro-ischemic influences, such as ‘Virchow’s Triad’: myocardial infarction-associated endothelial injury from blood stasis due to thrombus-forming ventricular dysfunction, and hypercoagulability induced by inflammatory mediators—in particular ROS and inflammatory cytokines (Virchow’s Triad) [98]. In addition to their clinical use for thrombic disorders, blood thinning compounds are often added to the working fluid of some kinds of medical equipment such as heart–lung machines, blood transfusion bags, and dialysis apparatus [99,100].

Adverse effects of blood thinners. The most dangerous and frequently occurring side effect associated with the use of blood thinners is the induction of both major and nonmajor bleeding—the severity and threat to health for a patient being significantly dependent on the type of drug used, pre-existing health conditions, and age [101]. For example, the annual incidence of life-threatening bleeding events for persons administered warfarin, a commonly-used anticouagulant, was 1–3% annually as of 2011 [102,103]. A principal hazard associated with many of these agents is their vitamin K antagonism properties—which increases bleeding risk and has motivated the development of new, non-vitamin K antagonist oral anticoagulants which appear less prone to causing life-threatening bleeding events versus warfarin [104,105]. A particularly useful agent in this class is apixaban, marketed as Eliquis, which has an excellent safety and efficacy profile and has emerged as a primary choice of caregivers for antithrombotic therapy [103,106]. Apixaban has nevertheless been shown to carry some risk of bleeding events as revealed by comparisons with edoxaban, reported in 2022 [107], and a 2017 study showing that this occurred in 107 people administered the drug, per 1000 in the study population, with minor bleeding in 167 per 1000 subjects [108].

### 3.4. Aspirin

This compound, formed by the esterification of salacylic acid to acetylsalicylic acid (ASA), is a naturally occurring component of willow bark which has extensive historical use by humans as an analgesic in the form of bark extracts and tinctures—and is currently marketed without requirement for medical prescription in purified form as a nonsteroidal anti-inflammatory drug (NSAID) used to suppress fever, pain, and other inflammation-associated aspects of a wide range of illness and trauma—with corollary use as an antithrombitic [109,110]. There is also some evidence for the value of this agent in decreasing the risk of ischemic stroke and secondary heart attack in susceptible individuals [111,112]; nevertheless, support for use of regular aspirin intake as general prophylaxis against CVD is substantially anecdotal at the time of this writing [113,114].

Adverse effects of Aspirin. Aspirin is sold in most countries as a generic drug, generally regarded as safe (GRAS) due to its long historical usage record in many societies. Nevertheless, some adverse side effects of the agent are known—and it should therefore be used with caution appropriate to the presence of risk factors for a particular patient, with awareness of the attendant drawbacks. For example, the compound is known to cause gastrointestinal irritation, with resulting stomach upset [112]. For some persons, this side effect may become severe, resulting in the development of bleeding from the stomach and, occasionally, ulcers [112,115]. Other adverse outcomes of aspirin use include increased severity of respiratory symptoms in asthma patients and exacerbation of bleeding by patients being treated with blood thinner regimens [116]. Additionally, high aspirin doses may result in tinnitus (ringing in the ears) and may increase risk of Reye syndrome in children [117].

## 4. Emerging Plant Phytochemical Supplements with Cardiovascular Applications and Functional Foods

### 4.1. Ginkgolides

Ginkgolides are phytochemical compounds present in the leaves, roots, fruit, and bark of the Ginkgo tree, the oldest living survivor of its taxonomic order: Ginkgolides, with its origins as a distinct class of plant species dating from over 290 million years ago. The plant is a native of China, where its fruit is consumed—and other parts of the plant have been used in traditional medicine through the history of Chinese societies, with widespread usage documented from the 11th century C.E [118,119]. Anecdotal but well documented descriptions of this use describe it as showing benefit in the treatment of kidney and bladder disorders, bronchitis, and asthma—with numerous reports suggesting its value in the treatment of dementia [118,119]. An analysis of these reports by the European Medicines Agency Committee on Herbal Medicinal Products in the context of modern understanding of natural medical materials (NMM) concluded that extracts of ginkgo leaf may be of mild benefit for peripheral vascular disease and dementia in elderly patients [120,121]. Nevertheless, investigations during the late 1900s and the early part of this century, failed to reveal the potent capacity of the plant extracts to prevent or remediate any serious disease. This notwithstanding, (weak) preliminary evidence exists for the ability of ginkgo to at least mildly ameliorate major symptoms of dementia and tardive dyskinesia symptoms in schizophrenia patients [121,122].

The discouraging outcomes of searches by other investigators for evidence of the clinical value of ginkgo components initially led our laboratory to exclude this NMM as a candidate for the development of novel strategies for the use of dietary and complementary medical approaches to counteract dysregulated inflammatory processes in disease—particularly CVD. Nevertheless, the biochemical nature of phytochemicals in the plant was sufficiently intriguing to lead members of our laboratory to speculate that one or more of them might exhibit potentiation of the cardioprotective effect of FK506, a macrolide immunosuppressant, which at the time of our research planning had been shown to inhibit cellular signaling processes in cardiac myocytes, known to be the underlying pathogenic mechanism of cardiac hypertrophy—as shown in Figure 2 below. Extracts of ginkgo leaves contain proanthocyanidins, bilobalides, phenolic acids, quercetin, flavonoid glycosides, such as kaempferol, isorhamnetin, myricetin, and terpene trilactones [chemical analysis and quality control] of *Ginkgo biloba* leaves, extracts, and phytopharmaceuticals [123,124,125]. The leaves also contain unique ginkgo polyphenols alkylphenols and biflavones [125]. This pharmacopia served as incentive to test the leaf extract (EGb761) for capacity to amplify the effects of FK506 in counteracting major consequences of cardiovascular I/R injury with a corollary effect of inhibiting the potentially pathologican pro-inflammatory activity of activated T lymphocytes—via the parallel signal modulation illustrated in Figure 2. The major clinical benefit being the potentiation of pharmacological effects and reduced dosage requirements for FK506—with resultant lowered toxicity in patients. The possibility of achieving such an effect was raised by the observation that terpene trilactone ginkgolides present at high concentrations in EGb761 exhibit dual ability as a potent antioxidant and platelet-activating factor (PAF) receptor antagonist, as shown below in Figure 2 [126].

The authors of the present report undertook the experiments previously described to test a hypothesis that combined therapy with macrolides, such as FK506 co-administered with ginkgolides, would reveal synergism in the cardioprotective capacity of these agents. Our results validate observations by other investigators demonstrating the cardioprotective properties of FK505 [127,129,130], and the discovery in our laboratory that combination therapy using FK506 plus EGb 761 synergistically augments postischemic cardiac function, while reducing the occurrence of reperfusion-induced arrhythmias [18], offering the potential for expanding the clinical utility of FK506 and allowing therapy with this drug and other macrolides—possibly including rapamycin, at lower doses than are currently useful [131].

CVD, cardiac surgery, and other heart trauma frequently result in periods of myocardial ischemia. The restoration of blood supply and reoxygenation of the myocardial tissue causes characteristic reperfusion injury, manifested by symptoms that include ventricular arrhythmias and deterioration in cardiac functions [132,133]. This effect is due substantially to myocardial membrane lipid oxidation by pro-inflammatory free radical species produced in a burst during the first minutes of reflow, resulting in impairment of the cell membranes [134,135]. Moreover, myocardial function may be improved by the administration of antioxidants following ischemic events [134,136,137]. Extending our hypothetical model at the time the above experiments were conducted, it was speculated that EGb 761 and selected component ginkgolides were expected to interact with FK506 so as to diminish reperfusion injury by two major mechanisms: one involving the quenching of reperfusion-associated free radicals and a second, dependent on their PAF-receptor antagonist properties—which limits cytoplasmic calcium increase in cardiomyocytes [138,139], thereby reducing the level of transcription factor nuclear factor activation of transcription (NF-AT)-dependent signals for the expression of pro-hypertrophic and pro-inflammatory genes in cells of the isolated hearts.

### 4.2. Heme Oxygenase Inducers

Heme oxygenases (HMOX) are heat shock protein (HSP) enzymes with a major physiological function of degrading heme released by red blood cell turnover to biliverdin and bilirubin; ferrous iron (Fe^2+^); and carbon monoxide (CO), as shown below in Figure 3. The potential toxicogenic effects of Fe^2+^ are counteracted through its sequestration to the ferritin protein, but the metal nevertheless remains capable of generating reactive oxygen species via Haber–Weiss or Fenton reactions [138,140]. Bilirubin and CO, produced as the end products of the HMOX degradation of heme, act as essential signaling molecules for a diverse array of essential physiologic processes—with bioactive properties that have become the focus of numerous efforts to prevent and treat chronic illnesses which have proven refractory to current therapeutic approaches [141,142].

## 5. Major Physiologic Effects of HMOX Product Metabolites

Carbon monoxide (CO): This signaling molecule stimulates cGMP-mediated vasodilation and suppresses ET-1—thus reducing the severity of hypertension—inhibits the proliferation of vascular smooth muscle cells (reducing need for stenting of cardiac patients); suppresses ventricular fibrillation and ischemia/reperfusion injury; and stabilizes Na^+^ and K^+^ distribution across membranes of cardiovascular cells [2,143].

Bilirubin: This potent endogenous antioxidant stabilizes Na^+^ and K^+^ distribution across membranes of cardiovascular cells and quenches ROS at levels that inhibit ischemia/reperfusion injury and diminish the occurrence of atherosclerosis and heart failure [2].

Three HMOX isoforms are encoded by most vertebrate genomes and expressed in response to epigenetic cues. These are as follows: Heme oxygenase 1 (commonly HO-1, aka HSP32), a stress-inducible form with immunoregulatory and cytoprotective roles which have made it the major focus of efforts to develop this enzyme class as clinical tools [144,145]. Heme oxygenase 2 (HO-2) is a constitutively expressed variant produced primarily in the brain, endothelial cells, gastrointestinal tract, and testes [144,146]. Moreover, whereas HO-1 is inducible by a wide variety of stressors and drugs, HO-2 is inducible only in response to adrenal glucocorticoids [147]. Heme oxygenase 3 (HO-3) is the third heme oxygenase variant, a 33 kDa protein expressed in kidneys, liver, and prostate [144,147] but with no clear biological role established for this isoform at the time of this writing.

In addition to its hemodynamic/housekeeping role, enzymes in this class serve as essential cellular cytoprotectants through their antioxidant capacity, protecting cells and tissues against oxidative damage by maintaining the tight regulation of inflammatory processes [2,148]. As understanding of the bioactive properties of this remarkable heat shock protein has increased—especially in the decade prior to this writing, the potential for the development of novel products and strategies based on its manipulation seemed enormous. A major advantage that HMOX-based interventions would have versus the currently available countermeasures to pathological inflammation, is a consequence of the fact that because exogenously administered antioxidants such as vitamins localize mainly to interstitial spaces, their clinical utility is limited—making them mostly unsuitable as first line countermeasures to pathologically activated inflammatory processes [128,149].

Exogenous antioxidants are typically small molecules obtained through diet, which decrease oxidative stress burden by the chemical inactivation of reactive oxygen metabolites. One well-known example is vitamin C (l-ascorbic acid or l-ascorbate), an aqueous reducing agent for oxygen radicals, which is, in turn, regenerated by glutathione-mediated reduction [128,150]. This vitamin is a major dietary antioxidant which additionally acts as a cofactor for antioxidant enzymes [128]. Vitamin E (tocopherols) are another major class of antioxidant vitamins. Tocopherols are lipid-soluble methylated phenols which reduce oxidized ascorbate thus augmenting vitamin C bioactivity [151,152]. In vertebrates, they act in concert with other small molecules such as ascorbate and the tripeptide glutathione, which includes a cysteine sulfhydryl group (-SH) that functions as a proton donor in the quenching of oxygen metabolites [153,154]. Several vertebrate enzymes degrade reactive oxygen species into molecules that are either harmless or rapidly metabolized. For example, superoxide dismutases convert superoxide into hydrogen peroxide (H_2_O_2_), oxygen, and hydrogen (McCord and Fridovich 1988)—a reaction occurring in tandem with the conversion of H_2_O_2_ into O_2_ and water by catalase [155,156], and peroxidase activity, which neutralizes organic hyperperoxides, such as the toxic products of lipid peroxidation, thereby supporting cell membrane integrity and the regulation of potentially pro-inflammatory and pathological cellular signaling processes [157]. Nevertheless, host antioxidant defenses in the form of endogenously produced compounds, such as glutathione, are typically overwhelmed by high levels of reactive oxygen species that occur as a major features of processes by which dysregulated inflammatory signaling leads to disease—including the critical properties of cancer and carcinogenesis [128,158]. By contrast, heme oxygenases perform their biological functions both at their sites of synthesis within cells, in which form they are termed intracellular HOs (iHOs), along with extracellular forms, (eHOs) which are distributed systemically in the circulation and other anatomical venues [128,158]. Moreover, in most reactions, HOs do not directly inactivate the cytotoxic properties of reactive oxygen molecules. Rather, their primary contribution to antioxidant defense occurs via the immunoregulatory effects of HMOX product metabolites CO and bilirubin—as previously described in Section 4 and Section 5 of this report, with a contributing discussion in Haines and Tosaki, 2018 [2,159]. Thus, HOs seemed to offer enormously valuable clinical tools which, due to their diverse venues for biological activity, avoid the limitations of exogenously administered drugs (including vitamins C, E, and others with antioxidant properties) which distribute to anatomical sites where their bioactivities are not major contributors to counteracting the underlying causes of inflammatory disease. Particularly intriguing findings have emerged from studies of heme oxygenase participation in cardiovascular physiology.

## 6. Heme Oxygenase in Cardiovascular Disease: Limits of Clinical Use

Ongoing studies of HO-1 biology are increasingly focused on the characterization of the enzyme’s capacity to mitigate the major symptom severity of cardiovascular disease [2]. The importance of these HOs to cardiac health is underscored by an emerging understanding of discoveries that CVD risk is increased significantly in persons with HO-1 gene promoter polymorphisms that adversely affect normal HO expression and regulation [160,161]. Moreover, since the 1980s, it is known that CO produced as a product of HO-1 heme degradation stimulates cardiovascular cells to augment vascular tone and vasodilation through guanylate cyclase stimulation and increased cGMP expression by vascular endothelial cells. These processes, which are physiologically coordinated with nitric oxide (NO) signaling, have been observed in hypertensive rodent models to result in increased coronary arterial pressure and blood flow while at the same time reducing total peripheral resistance [162,163]. These effects are reversed by the inhibition of HO-1 [163]. CO-induced guanylate cyclase activity, which increases cGMP-mediated events, has additionally been observed to inhibit the proliferation of vascular smooth muscle cells (SMC) with resulting occlusion of cardiac vasculature by neointima [163,164]. These observations offer the potential for novel strategies in the prevention of pathological vascular remodelling and for improved drug-eluting stent design. Additionally, bilirubin produced by HO-1 activity reduces platelet aggregation and leukocyte—processes that contribute to cardiovascular inflammation [163,164].

Both CO and bilirubin produced by HO-1 activity in the cells of cardiovascular tissue ameliorate a diverse range of cardiovascular pathologies due to the effect of these two degradation products on vascular tone [163,164]—along with the stabilization of K^+^ and Na^+^ ion membrane distribution [165]. Moreover, the regulatory effects of HO-1-generated bilirubin on the expression of ROS by NADPH oxidase has also been demonstrated to be cardioprotective as evidenced by the prolonged survival of mouse-to-rat cardiac xenografts, the inhibition of IR and balloon injury, and improved cardiac functions [163,166,167].

Related studies by the authors of this report and other investigators demonstrate HO-1 activity and enhanced carbon monoxide signaling induced by ischemia/reperfusion injury diminishes infarct size and enhances post-ischemic recovery in rat hearts ex vivo [168,169,170].

HO also exhibits potential for use as a countermeasure to cardiac hypertrophy, thereby suggesting that inducers of the enzyme may be useful in combination with FK506 and ginkgolides as described in the present report [18]. For example, an assessment of HO-1 as an antihypertrophic agent—published in 2009—demonstrated that an overexpression of HO-1 in cardiomyocytes in which NF-kB-dependent hypertrophy had been triggered via exposure to hydrogen peroxide caused the suppression of hypertrophy-associated cellular changes. This is a representative example of the plethora of exciting clinical applications of HO biology [171,172]. These encouraging research outcomes notwithstanding, by 2018, it had become evident that the development of novel HO-based strategies available to the public had fallen short of this apparent promise. Attempts to base workable clinical protocols on known properties of the enzyme encountered obstacles in the design of formulations capable of selectively increasing HO activities at sites optimal for preventing and treating disease [141,148]. In recognition of the above-mentioned constraints on the clinical use of HOs, the authors of this report undertook the characterization of contributions of these enzymes to cardiovascular function, and the remediation of CVD [2,4,173]. The above-described analyses of findings by other investigators was combined with in vitro, animal, and human clinical studies at the University of Debrecen and Kuwait University to evaluate clinical efficacy in phytochemical HO inducers—notably fruit seed kernel flavonoids—to counteract hyperinflammatory tissue damage in CVD and other disorders. This research initiative began in 2004, with the discovery by authors of this report of potent HO-inducing properties in the flavonoid fraction of seed kernels of sour cherry (*Prunus cerasus*) [165], and was extended in 2014 by a demonstration of the cardioprotective ability of sour cherry seed kernel flavonoids [174].

Thereafter, a major effort was undertaken by this laboratory to utilize HO inducers in sour cherry seed extracts (SCE) to mitigate the major symptoms of various serious chronic illnesses through the capacity of these enzymes to quench pathological ROS levels. The major focus of this research was in the realm of cardiovascular medicine. Representative studies in this area include the evaluation of endothelin-1 (ET-1) to attenuate hypertrophic signaling in H9c2 rat cardiomyoblasts in vitro, corresponding to in vivo processes underlying cardiac hypertrophy [159]; the treatment of rat hearts ex vivo, with alpha-melanocyte-stimulating hormone (alpha-MSH) to induce vasodilation and mediate cardioprotection via an HO-dependent pathway [175]; and a demonstration in rats of beta carotene (BC)-mediated reduction in infarct size, increased total antioxidant capacity of the myocardium, and other cardioprotective effects occurring at relatively low BC dose but (counterintuitively) diminished at higher doses of BC—an effect which may be due to the inefficient clearance of Fe^2+^ produced by HO activity [176].

## 7. ‘Biotherapeutic’ Use of Heme Oxygenase in Osteoarthritis Treatment: Phase 1 Human Trials

Particularly encouraging evidence that the SCE flavonoid-mediated induction of HO activity may be a breakthrough means for the medical use of HO was offered by a human clinical study published in 2014 by authors of this report, showing that a topical SCE preparation dramatically reduced pain and inflammation-associated serum and immune parameters through systemic HO activation in a cohort of Kuwaiti osteoarthritis (OA) patients [177].

Figure 4 shows the major known pathomechanisms for OA. Potentially destructive lymphocyte activation and the resultant production of inflammatory mediators served as major therapeutic targets for the human study described in Mahmoud et al., 2014. As described by that report, the systemic SCE-mediated induction of HOs diminished pathological T cell activation, resulting in significantly decreased pain, along with a significant reduction in the serum content of C-reactive protein (CRP) [177]. The outcomes of this clinical investigation and related studies suggest that the capacity of sour cherry seed flavonoids to potently remediate pain and other symptoms of serious if dysregulated inflammatory processes—as described—finally provides a means to overcome obstacles to the wide clinical use of heme oxygenases—reviewed by Hopper et al., 2018 [141].

## 8. Dietary Phytochemicals/Functional Foods

Almost all foods alter an individual’s risk for and response to disease and those rich in plant phytochemicals may be particularly beneficial for CVD prevention and management. Foods selected and blended to ‘support’ endogenous countermeasures to ill health are known as “functional foods”, a term that encompasses a wide range of substances regularly consumed as components of the human diet. As discussed in the first part of Section 2 of this report, supplements, which may include phytochemical-rich edible material such as spices like turmeric and nutmeg, capable of increasing the expression of the endogenous antioxidant enzyme heme oxygenase and extract of sour cherry seeds, which also stimulates high expression of the enzyme, are classed as functional foods [55,178]. A particularly valuable outcome of the experiments on which this report is based, is the creation of an easy-to-use in vitro platform, by which functional food preparations (in the form of Kancor Aqua Spices and other foods containing “natural medical material” (NMM) may be used as aqueous phase reagents—and be assessed for synergism in their capacity to down-regulate potentially pathological signaling processes in human cells.

Dietary modifications to reduce CVD susceptibility and treat its symptoms are strategies most completely under the control of individuals at risk for developing these diseases; however, they are also among the most difficult to pursue successfully. Adherence to heart-healthy diets at levels sufficient to result in a significantly positive impact on cardiovascular health typically requires a level of commitment and sustained discipline that many people are incapable of, even with the knowledge that failure to do so puts their life in jeopardy. Moreover, highly specialized dietary regimens, with attendant involvement of medical caregivers and paid advisers, often require financial outlays that exceed the resources most people have to draw on. The investigation on which this report is based was conducted to aid in meeting this challenge. The data shown here offer an approach by which optimal combinations of selected dietary components with known benefits in stabilizing disease-associated cellular signaling pathways may be rapidly identified—leading to food and flavoring compositions with preventive and therapeutic potency in the same range as pharmaceutical drugs. First, some major heart-healthy foods should be considered.

Most people have some insight into aspects of diet and lifestyle contributing to generally good health—including the avoidance and management of CVD. This includes commonsense behavior such as regular exercise and the avoidance of smoking and alcohol. Moreover, in the latter part of the 1900s, continuing into this century, scientific understanding and public education have revealed a correlation between certain food groups and various aspects of health. For example, outcomes of a meta-analysis published in 2014 provided comprehensive descriptions of how fruits and vegetables decrease the CVD risk and mortality [179]. Related studies demonstrated vegan diets to correlate with better cardio-metabolic profiles versus omnivorous food intake [180,181]. Interestingly, the Mediterranean diet, which is high in fats, is more effective in controlling cholesterol and high blood pressure than ‘generic’ low fat diets [182,183].

Some foods contain phytochemicals that are particularly effective at restoring tight physiologic regulation to pro-inflammatory cellular signaling that becomes hyperactivated in all diseases at some level, in particular cardiovascular ischemia/reperfusion pathologies. As described above in this report, four spice flavorings are rich in these compounds and were selected as representative foods to demonstrate the capacity of the in vitro model used here to identify combinations of each of the four choices that exhibit synergism in the suppression of inflammatory signaling predictive of potential for the prevention and management of a wide range of inflammatory tissue injury, including those forms characteristic of ischemic disease.

## 9. Future Directions

A major emergent theme in the ongoing efforts to characterize the underlying molecular biological contributors to CVD and develop low cost strategies for preventive medicine and therapy is the increasing recognition of synergism as a major tool for augmenting the clinical efficacy and reducing the toxicity of drugs currently in use for both human and veterinary medicine. A paradigm of this broad strategy is provided by observations that terpene trilactone components of EGB761, an extract of *Ginkgo biloba* leaves, augmented the cardioprotective capacity of FK506, a macrolide immunosuppressant, and lowered the dosage of the drug needed to inhibit potentially pathological levels of inflammatory cytokine expression by ischemic/reperfused heart tissue in vivo [18,20]. In in vitro studies using a human T cell model, EGB761 was further shown to synergize with cyclosporine—another widely used but toxic immunosuppressant, in reducing the production of inflammatory cytokines [3,6]. The molecular mechanism by which ginkgo terpenoids interacted with both FK506 and cyclosporine to lower the drug dosage required for outcomes predictive of clinical value was due to the modulation of calcium signaling resulting from the engagement of PAF receptors on the surface of cardiac cells and T lymphocytes to affect calcineurin-mediated transcription factor translocation into the nucleus with improved efficiency and at drug lower dosages [3,6,18,21]. Consequently, ongoing investigations have focused on this molecular cascade. Specifically, candidate GRAS compounds—with priority given to components of the human diet—are being evaluated for their capacity to modulate calcium–calcineurin signaling with greater efficiency than ginkgolides. The preliminary results of this work have revealed highly intriguing evidence of the synergistic suppression of inflammatory cytokine expression by human cells commonly used spice flavorings. Experiments to fully characterize this effect are ongoing. Results at the time of this writing are particularly encouraging since the robust suppression of inflammatory tissue damage has been demonstrated for turmeric [46], ginger [47,48], capsicum/capsaicin—in red pepper [49,50], paprika [51,52], and nutmeg [53,54,55] as described in Section 2 of this report. In summary, the research described here describes the current understanding of cardiovascular diseases and approaches to its management, and explores novel strategies for the use of GRAS natural medical materials to potentiate clinical efficacy and reduce the toxicity of currently used therapies for CVDs.

## 10. Conclusions

Cardiovascular diseases have been primarily characterized by the dysfunction of the myocardium as the primary and major cause of morbidity and mortality and have shown an increased rate across the globe. Both experimental and clinical studies show that the activation of inflammatory responses, myocardial necrotic, apoptotic, and autophagic processes substantially aggravate cardiac dysfunction, leading to the development of heart failure, ventricular arrhythmias, and sudden cardiac death.

Many studies described that phytochemical components originated from vegetables, seeds, fruits, and different parts of trees, e.g., leaves and roots, are having several therapeutic applications for various diseases based on their action mechanisms in both animal and human subjects focusing on cardiovascular diseases [184]. We also discuss in the present review how the natural compounds modulate their molecular targets and signal transduction mechanisms in cardiovascular diseases including inflammation ischemia/reperfusion-induced injury as oxidative mediated pathogenesis, providing indications for their use in further clinical trials.

The use of phytochemical ingredients based on mostly classic and traditional Chinese medicines for the treatment and prevention of cardiovascular diseases are gradually being introduced nowadays.

Finally, it is of interest to note that phytochemicals, although it was not the theme of the current review, are being found to have substantial epigenetic potential in modifying gene expressions, resulting in the inhibition of the spreading of various tumors [185]. The epigenetic effects of phytochemicals need further and controlled studies in tumor therapies under both experimental and clinical conditions.

## Figures and Tables

**Figure 1 ijms-25-06176-f001:**
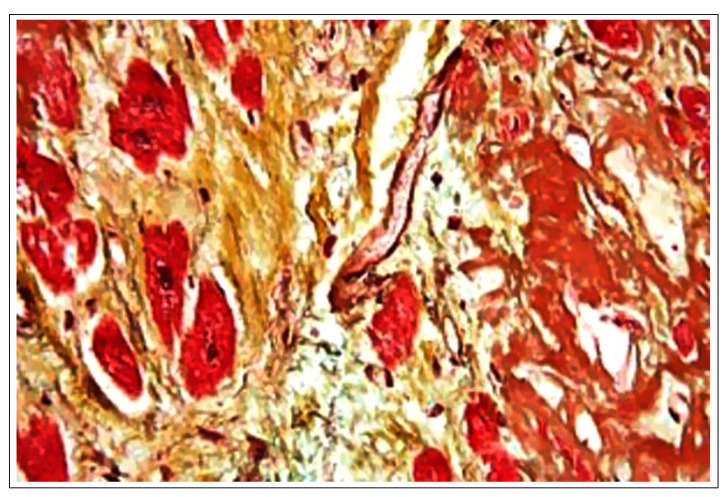
High magnification micrograph of senile cardiac amyloidosis. Movat stain (black = nuclei, elastic fibres; yellow = collagen, reticular fibers; blue = ground substance, mucin; bright red = fibrin; red = muscle). Human autopsy specimen. The micrograph shows amyloid (extracellular muddy brown material—right of image), abundant lipofuscin (dark red granular material) and myocardial fibrosis (yellow—left of image). Related images Intermed. mag. (H&E). High mag. (H&E). Very high mag. (H&E). Intermed. mag. (congo red). High mag. (congo red). Very high mag. (congo red). Very high mag. (Movat’s stain). SOURCE: Wikimedia Commons, Copyright © 2011 Michael Bonert (https://en.wikipedia.org/wiki/Cardiovascular_disease#/media/File:Cardiac_amyloidosis_very_high_mag_movat.jpg (accessed on 16 May 2024)).

**Figure 2 ijms-25-06176-f002:**
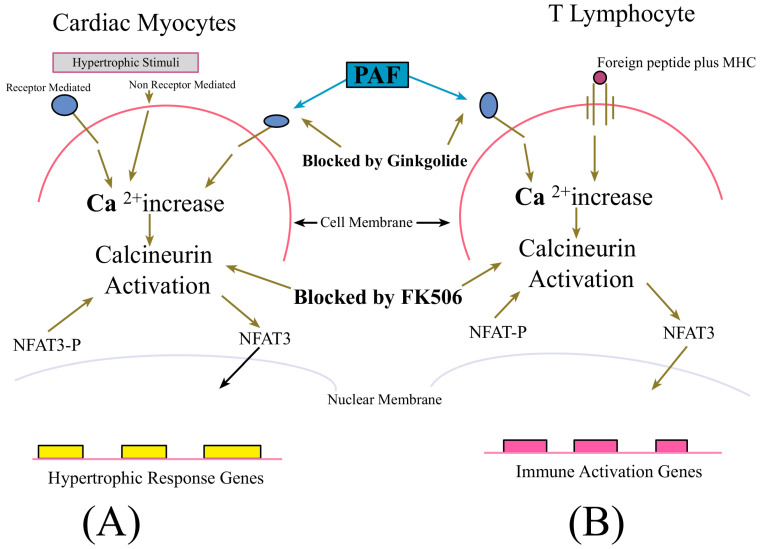
Parallel regulation of intracellular calcium-mediated hypertrophic/arrhythmogenic stimulus in cardiac myocytes and immune activation in T lymphocytes using FK506, and PAF-receptor-inhibitory ginkgolides from leaf extract (EGb761). In cardiac myocytes, the mechanical stimulation of stretch receptors and macromolecular stimulation of cell membrane-embedded Ca^2+^ channels increase cytoplasmic Ca^2+^ levels, causing calcineurin phosphatase-mediated linearization of NFAT transcription factors with resulting expression of genes contributing to hypertrophy and arrhythmogenesis [127]. An analogous process results in the activation of T lymphocytes, engendering pro-inflammatory reactions that may become pathological [128].

**Figure 3 ijms-25-06176-f003:**
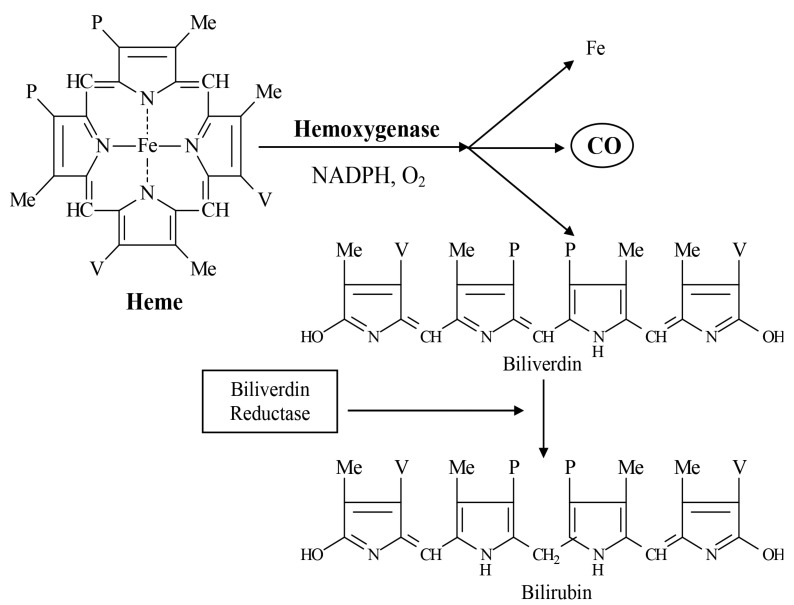
Heme oxygenase activity: reaction stoichiometry and physiological effects of product metabolites. Heme oxygenases degrade heme by cleavage of the heme ring at its alpha-methene bridge, producing carbon monoxide (CO), ferrous ion (Fe^2+^), and biliverdin, which is further reacted by biliverdin reductase to bilirubin (a major endogenous antioxidant).

**Figure 4 ijms-25-06176-f004:**
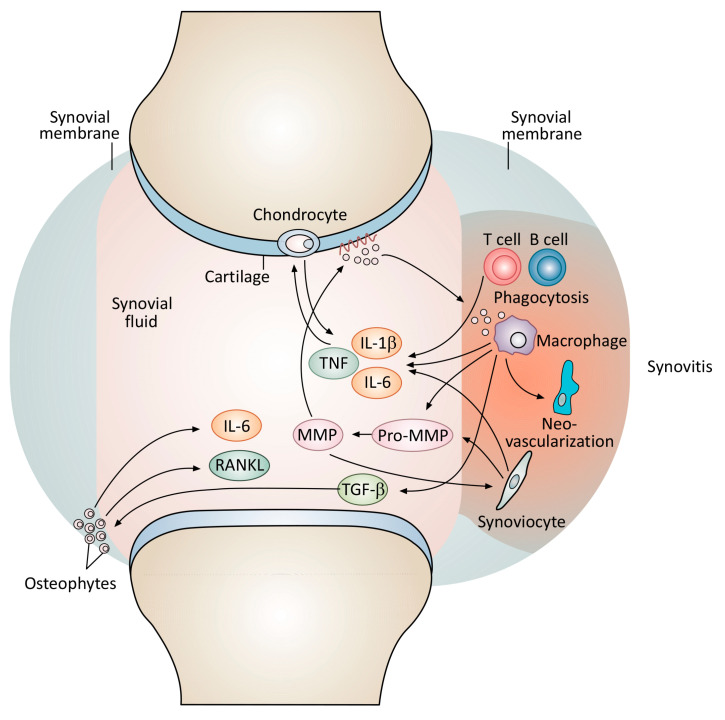
Major pathomechanisms contributing to OA-associated articular tissue damage. Physical trauma and reactive oxygen species produced as an adaptive response by host cells increase T cell, macrophage, and synoviocyte production of pro-inflammatory cytokines, including interleukin-6 (IL-6), TNF-A, and interleukin1-beta (IL1-B). From Mahmoud FF, Al-Awadhi AM, Haines DD. Amelioration of human osteoarthritis symptoms with topical ‘biotherapeutics’: a phase I human trial. Adapted with permission from Ref. [177].

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
