# Peer review of "Evolving Strategies for Use of Phytochemicals in Prevention and Long-Term Management of Cardiovascular Diseases (CVD)"

_ijms, 2024, doi:10.3390/ijms25116176_

Round 1

Reviewer 1 Report

Comments and Suggestions for Authors

The present manuscript describes current and novel strategies for prevent cardiovascular disease (CVD).

The manuscript is well written, the language used is correct throughout the text and the references are appropriate, adequate, and up-to-date.

However, us currently known inflammation and oxidative stress represent the two major pathways of endothelial dysfunction. The authors are encouraged to report in 3-4 lines-sentences, before "future directions", from current and existing literature some novel biomarkers of micro-inflammation and oxidative stress, and extracellular matrix degradation proteases and native tissue inhibitors in order to associate them with the mentioned above madiated cascades of signaling from cell surface receptors of cardiovascular and immune cells with protective and pathological epigenetic features, in order to better and more precisely orient future directions focusing on novel diagnostic and therapeutic approaches. 

Author Response

The authors thank to the reviewer for her/his valuable and very kind comments that she/he made about our manuscript.The revision of our manuscript has been carried out, including the suggestions and comments of Reviewers Number 1 and 2, and we apologize for any confusion, which may have been initially arisen. Additionally, several new references have been also incorporated in the revised version of the manuscript, as the reviewers 1 and 2 suggested. The authors believe that the manuscript has been substantially improved. All changes are written in red in the revised version.

Reviewer 2 Report

Comments and Suggestions for Authors

This report discusses the primary pathomechanisms of disease focusing on dysregulated inflammatory processes, particularly in cardiovascular disease (CVD), and explores strategies for counteraction through the synergistic interaction of various agents, including pharmaceutical drugs and generally regarded as safe (GRAS) natural medical materials. Congratulations on the authors on the comprehensive review

Few comments below

  • In the introduction, clearly outline the objectives and scope of the review. The introduction does not seem to explain the provide a real introduction to the topic and reads as a separate mini review
  • Line 16- “The 15 well-defined CVD 16 classes are explored with particular emphasis on the extent to which oxidative stressors and associated ischemia-reperfusion tissue injury contribute to major symptoms” – What do you mean by 15 well-defined CVD classes?
  • Consider adding a conclusion section with main/key points discussed in the review, before adding future directions
  • Future directions- consider expanding the added value of this research to the current clinical practice and guidelines of management of CVD

Author Response

The authors thank this Reviewer (Number 2.) for her/his valuable comments that she/he made about our manuscript. The revision of our manuscript has been done, including the suggestions and comments together with those of Reviewer Number 1., and we apologize for any confusion, which may have been initially arisen. Several new references have been incorporated in the revision. All changes have been marked in red in the revised version.

Introduction: In pages 5-6: New references (from 27 to 32) and explanations have been added to the revised version of the manuscript, which may give new insights to the topic of this review.

“This work was the basis for Multi-Threat Medical Countermeasures (MTMC) technology which aimed to develop treatments for pathologies triggered by multiple classes of chemical agents with a single drug protocol (Cowan et al. 2004, 2006 and unpublished observations) (27). MTMC technology may also find more general public heath applications because of the commonality of inflammatory mechanism in chemical insult and disease (27-28). MTMC synergistic combinations of anti-inflammatory drugs and/or phytochemicals have been confirmed in assay and animal models (29-30). Further drugs with MTMC primary or pleotropic anti-inflammatory pharmacology are prescribed and administer to millions of patients often in polypharmacy with potential synergy and implications for design of MTMC polypills (30). MTMC can be generic drug or supplements addressing affordability and practiced in any small clinic reducing R&D costs. MTMC synergy assay and animal models for inflammation are readily available (26, 31). GLP-1 diet drugs are FDA approved for cardiovascular disease and have demonstrated efficacy for a myriad other inflammatory pathology (32). Concerns of cost for on-patent MTMC vs more affordable generic MTMC have been considered (28).”

Line 16- (in the first version).

These categories of CVD include the various types of angina (e.g., angina pectoris, stable and unstable angina), ischemia-induced arrhythmias, reperfusion-induced arrhythmias, coronary sclerosis, myocardial infarction and related consequences, e.g., high blood pressure, genetically originated arrhythmias (e.g., Brugada syndrome, long QT syndromes, PVC, Wolf-Parkinson-White syndrome, “torsade de pointes”) arrhythmias. Please, see the publication by Tosaki A., 2020, ref: 133 in the revised version, Front Pharmacol 2020, 11, 616, doi:10.3389/fphar.2020.00616).

Conclusion and future direction – Beside the treatment of cardiovascular diseases, new therapeutic indication of phytotherapy is incorporated (e.g., tumor therapy) in the Conclusion (see below, and in red in the revised version).

“Cardiovascular diseases have been primarily characterized by the dysfunction of the myocardium as the primarily and major cause of morbidity and mortality, and showing an increased rate, in the Globe. Both experimental and clinical studies show that activation of inflammatory responses, myocardial necrotic, apoptotic and autophagic processes substantially aggravate cardiac dysfunction, leading to the development of heart failure, ventricular arrhythmias and sudden cardiac death. Many studies described that phytochemical components originated from vegetables, seeds, fruits and different parts of trees e.g., leaves and roots, are having several therapeutic applications for various diseases based on their action mechanisms in both animal and human subjects focusing on cardiovascular diseases (Guo B et al., 2024, Biomed Pharmacotherapy, 2024doi: 10.1016/j.biopha.2024116313). We also discussed in the present review how the natural compounds modulate their molecular targets and signal transduction mechanisms in cardiovascular diseases including inflammation ischemia/reperfusion-induced injury as oxidative mediated pathogenesis, providing indications for their use in further clinical trials. The use of phytochemical ingredients based on mostly classic and traditional Chinese medicines for the treatment and prevention of cardiovascular diseases are gradually being introduced nowadays. Finally, it is of interest to note that phytochemicals, although it was not the theme of the current review, are having substantial epigenetic potentials and modifying gene expressions resulting in the inhibition of the spreading of various tumors (Rajendran P et al., Int J Mol Sciences, 2022, doi: 10.3390/ijms231911712). The epigenetic effects of phytochemicals need further and controlled studies in tumor therapies in both under experimental and clinical conditions.”

Round 2

Reviewer 2 Report

Comments and Suggestions for Authors

The authors have address the comments at revision and have significantly improved the manuscript